# Microbiological Air Quality in Heating, Ventilation and Air Conditioning Systems of Surgical and Intensive Care Areas: The Application of a Disinfection Procedure for Dehumidification Devices

**DOI:** 10.3390/pathogens8010008

**Published:** 2019-01-15

**Authors:** Michele Totaro, Anna Laura Costa, Beatrice Casini, Sara Profeti, Antonio Gallo, Lorenzo Frendo, Andrea Porretta, Paola Valentini, Gaetano Privitera, Angelo Baggiani

**Affiliations:** 1Department of Translational Research and New Technologies in Medicine and Surgery, University of Pisa, Via San Zeno 35-39, 56123 Pisa, Italy; micheleto@hotmail.it (M.T.); alauracosta@alice.it (A.L.C.); beatrice.casini@med.unipi.it (B.C.); profeti.sara@gmail.com (S.P.); lorenzo.frendo@hotmail.com (L.F.); andrea.porretta@med.unipi.it (A.P.); paola.valentini@unipi.it (P.V.); gaetano.privitera@med.unipi.it (G.P.); 2Department of Public Health and Hygiene, Azienda USL Toscana Nord Ovest, 56100 Pisa, Italy; antonio.gallo@uslnordovest.toscana.it

**Keywords:** HVAC, *Aspergillus* spp., hydrogen peroxide, air disinfection

## Abstract

International literature data report that the increase of infectious risk may be due to heating, ventilation and air conditioning (HVAC) systems contaminated by airborne pathogens. Moreover, the presence of complex rotating dehumidification wheels (RDWs) may complicate the cleaning and disinfection procedures of the HVAC systems. We evaluated the efficacy of a disinfection strategy applied to the RDW of two hospitals’ HVAC systems. Hospitals have four RDW systems related to the surgical areas (SA1 and SA2) and to the intensive and sub-intensive care (IC and sIC) units. Microbiological air and surface analyses were performed in HVAC systems, before and after the disinfection treatment. Hydrogen peroxide (12%) with silver ions (10 mg/L) was aerosolized in all the air sampling points, located close to the RDW device. After the air disinfection procedure, reductions of total microbial counts at 22 °C and molds were achieved in SA2 and IC HVAC systems. An *Aspergillus fumigatus* contamination (6 CFU/500 L), detected in one air sample collected in the IC HVAC system, was eradicated after the disinfection. The surface samples proved to be of good microbiological quality. The results suggest the need for a disinfection procedure to improve the microbiological quality of the complex HVAC systems, mostly in surgical and intensive care areas.

## 1. Introduction

Several studies report the lack of air quality control in operating rooms as a key factor for surgical site infections following the most common general surgery procedures [1,2]. Surgical site infection accounts for 13%–17% of the total amount of nosocomial infections. Air quality control is therefore routinely performed in surgical settings [3,4,5] according to procedures described by international standards [6,7]. To achieve ultraclean air circulation in operating rooms, Italian standards highlight the technical requirements for the Heating, Ventilation and Air Conditioning (HVAC) plants [8,9]. Moreover, ISO 14644-1 [7] defines indications for the planning of HVAC systems and for the management of air quality in surgical areas. Despite this standard being aimed at preventing the occurrence of contamination and infection in patients hosted in the high-risk areas of hospitals, several research studies suggest that some HVAC systems may be a source of pathogens, because humidifiers, dirty air ducts and filters could be the perfect sites for the growth and dissemination of indoor bacteria and molds [10,11]. On the other hand, new updated studies state that an improvement in the microbiological quality of air entering surgical areas may be achieved by low-turbulence airflows [12], and by the introduction of new devices and construction materials [13]. In fact, ISO 16890 [14] describes an efficiency classification system of air filters for general HVAC plants. This standard provides an overview of the test procedures and specifies the general requirements for assessing and marking HEPA filters, as well as for documenting test results. Moreover, HVAC systems may include special devices to improve indoor air quality and save energy [15]. In this study, we describe a microbiological risk management scheme applied to the HVAC systems in two Italian hospitals. We evaluated the disinfection strategy efficacy in hospital HVAC systems with complex dehumidifier devices.

## 2. Results

### 2.1. HVAC Systems in Surgical Area 1 (SA1)

From the SA1s of both hospitals, the microbiological parameter values were always within the limits provided by the Italian guidelines [6]. In detail, before chemical disinfection, the total microbial count of the air samples taken at 22 °C and 37 °C ranged from 13 to 2 CFU/500 L, while the mold count observed ranged from 5 to 2 CFU/500 L (Figure 1).

Considering the low contamination state detected before the disinfection, we did not observe a significant reduction in contamination in both hospitals after the treatment (*p* = 0.052; *p* = 0.058). In fact, 24 h after the disinfection, all the values were always below 5 CFU/500 L. Moreover, no differences were detected between the various sampling points. The total microbial count of the surface samples and the mold levels were always below 1 CFU/dm^2^. This result may be due to the high temperatures of the analyzed surfaces. 

### 2.2. HVAC Systems in Surgical Area (SA2)

In both hospitals, all the microbiological data results were within the limits provided by the Italian guidelines. On the analyzed surfaces, a low contamination level was detected, with counts below 1 CFU/dm^2^.

The statistically significant reduction of the total microbial count, at 22 °C, and the mold level was detected in both hospitals. In fact, the disinfection treatment reduced the air bacterial counts from 39 ± 6 and 65 ± 13 CFU/500 L to 7 ± 5 and 11 ± 7 CFU/500 L in Hospitals 1 and 2, respectively (*p* = 0.011; *p* = 0.009) (Figure 2). A reduction in the mold level was achieved in Hospital 1 (from 18 ± 8 to 7 ± 4 CFU/500 L) and in Hospital 2 (from 15 ± 6 to 9 ± 7 CFU/500 L) (*p* = 0.046; *p* = 0.048).

No air contamination differences were observed between the various sampling points. 

### 2.3. HVAC Systems in the Intensive Care (IC) Unit

All the microbiological data results were within the limits of the Italian guidelines. Once again, from the surface samples, low contamination levels were observed, with counts below 1 CFU/dm^2^.

From the intensive care wards of both hospitals, a statistically significant reduction of the total microbial count in the air samples, taken at 22 °C, and the mold level was detected. In detail, disinfection with hydrogen peroxide reduced the total microbial count, at 22 °C, from 17 ± 4 and 11 ± 3 CFU/500 L to 4 ± 3 and 4 ± 2 CFU/500 L in Hospitals 1 and 2, respectively (*p* = 0.013; *p* = 0.021) (Figure 3). 

Mold reduction was achieved in Hospital 1 (from 16 ± 4 to 0 CFU/500 L) and in Hospital 2 (from 5 ± 1 to 0 CFU/500 L) (*p* < 0.001).

Before the disinfection treatment, in Hospital 1, an *Aspergillus fumigatus* contamination (6 UFC/500 L) was detected and isolated from point D (room vent). After the hydrogen peroxide aerosolization, all the molds were eradicated from the HVAC system.

### 2.4. HVAC Systems in the Sub-Intensive (sIC) Unit 

In the sub-intensive care wards, the microbiological results obtained from the air and surface samples were within the limits of the Italian guidelines. The surface samples had total microbial and mold counts below 1 CFU/dm^2^.

The disinfection treatment reduced the total microbial count in the air at 22 °C from 21 ± 5 and 6 ± 5 CFU/500 L to 5 ± 3 and 3 ± 2 CFU/500 L in Hospitals 1 and 2, respectively (*p* = 0.032; *p* = 0.046) (Figure 4).

The reduction of the total microbial count at 37 °C (from 5 ± 3 to 0 CFU/500 L and from 2 ± 1 to 0 CFU/500 L) was not statistically significant (*p* = 0.051; *p* = 0.059). Similarly, the mold reduction (from 5 ± 4 to 0 CFU/500 L and from 2 ± 1 to 0 CFU/500 L) was not statistically significant (*p* = 0.053; *p* = 0.051). 

No air contamination differences were observed between the various sampling points. 

## 3. Discussion

The literature reports that almost $10 billion is spent annually on hospital-acquired infections, with surgical sites accounting for 34% of the overall cost. Furthermore, the air quality in operating rooms is an important factor that may contribute to surgical site infections [16,17]. The use of HVAC systems in healthcare buildings is important to ensure the circulation of clean air and in order to prevent hospital infections, but at the same time, they require significant amounts of energy to operate [18]. 

Considering that the use of HVAC systems in the operating rooms and intensive care areas of hospitals has increased worldwide, the literature data suggest the need for new devices associated with HVAC systems aimed at saving energy costs [15]. New technical devices included in the HVAC systems are often large and complex and their cleaning and disinfection is difficult to achieve. In our study, the installation of rotating dehumidification wheels (RDWs) in four out of 90 HVAC systems has been included in the energy saving management plan of the hospitals. Despite the fact that the RDW devices allow for air dehumidification and the high temperatures prevent occurrences of air contamination, our experience reports the presence of *Aspergillus fumigatus* in the room vent of one hospital intensive care area. *Aspergillus* spp. are ubiquitous thermotolerant molds, which may disperse into air currents and deposit into human alveoli causing syndromes such as allergic bronchopulmonary aspergillosis, found mostly in immunocompromised patients hosted in the high-risk areas of hospitals [19]. Outbreaks of invasive *Aspergillus fumigatus* infection in several surgical and intensive care settings, associated with contaminated HVAC systems, are reported in many recent and old studies [20,21,22,23], which assert that mold contaminations were due to the lack of plant maintenance and the structural complexity of the HVAC system. 

Our research shows that contaminations of *Aspergillus fumigatus* and further filamentous fungi (data not shown) were eradicated after the introduction of the disinfection procedure with hydrogen peroxide applied to the RDW devices. This result is frequently associated with statistically significant reductions of the total microbial count at 22 °C. Before disinfection treatments, in all of the investigated HVAC systems, we detected total microbial counts at 22 °C to be significantly higher than the growth counts at 37 °C (*p* = 0.012; *p* = 0.011; *p* = 0.031; *p* = 0.030; *p* = 0.023; *p* = 0.030; *p* = 0.028; *p* = 0.025). These data assert the prevalence of environmental bacteria contamination in the absence of disinfection procedures. In fact, considering the low levels of mesophilic bacteria contamination, we have not observed statistically significant reductions of the total microbial counts, at 37 °C, after disinfection.

It is known that aerosolized hydrogen peroxide is often used for hospital indoor air disinfection, preventing outbreaks of multidrug-resistant bacteria and mold infections [24,25]. Hydrogen peroxide (HP) is a strong oxidizer, bactericidal at 3% solution, a sterilant at 6% with six hours of exposure, more powerful than chlorine dioxide, and more stable at high temperatures and high pHs compared to chlorine-based disinfectants. It is non-toxic to humans and it cannot damage several types of technical materials. The lack of toxicity of HP to people and animals and its lack of environmental impact have been confirmed by the U.S. Food and Drug Administration (FDA) and the U.S. Environmental Protection Agency [26,27]. 

## 4. Materials and Methods

### Hospital Settings

The healthcare settings (Hospital 1 and Hospital 2) are two general hospitals in the North-Western Tuscany local health unit (Italy), hospitals with 386 and 360 beds and with catchment areas of about 165,000 and 140,000 inhabitants, respectively. Hospital 1 has been active since 2014, while Hospital 2 has been active since 2016. Both architectural structures are monoblocks with a central plate on 5 levels. The warehouses and car parks are in the basement. The ground floor houses the emergency department. The medical and clinical areas are located on the first and second floor, while the surgery areas are on the third floor. 

### HVAC Systems

Each hospital presents 90 HVAC systems located on the hospital rooftop. They have the following devices (Figure 5): outdoor air intake dampers, pre-filtration and filtration systems, air recirculation devices, heating batteries, humidifiers and dehumidifier-cooling devices, and air transport channels inside the wards (with terminal HEPA filters).

Only 4 out of the 90 HVAC systems have a rotating dehumidification wheel (RDW), whose aim is to remove water vapor from the air and save energy during air recirculation. The RDWs have a temperature range, while operating between 70 and 100 °C. 

RDW devices are made of silicon gel (82%), glass fiber (16%) and acrylic coating (2%). As shown in Figure 6, the surfaces present a honeycomb shape, allowing flow air impact. The device allows air input (98% of relative humidity (RH); 10°C), which passes through it. Dehumidified air (0% of RH; 25 °C) enters into the channels and is subsequently recirculated (50% of RH; 20°C) entering into the RDWs. 

The RDWs are present in the HVAC systems related to the two surgery areas (SA1 and SA2), the intensive care area (IC) and the sub-intensive care (sIC).

### Air and Surface Sampling

From September 2016 to July 2018, in both hospitals, microbiological tests were performed on the four HVAC systems with RDW devices (Table 1). 

The same analyses were applied before and after the disinfection procedures.

From each HVAC system, air microbiological sampling was carried out using Microflow 90 (Aquaria, Italy) during the HVAC system activities. From the points shown in Figure 6 (air input—point A; air output—point B; recirculation—point C; room—point D), 500 L of air (flow rate of 120 L/min) was aspired for the research of molds and total microbial counts at 22 and 37 °C. 

For all the investigated HVAC systems, the surface microbiological sampling was performed in both RDW frontages and in the channels carrying the air into the rooms. Microbiological sampling on surfaces was performed using Contact Agar Plates (VWR, Italy) for the research of molds and total microbial counts at 22 and 37 °C.

The culture media used for the mold and total microbial count detection were Sabouraud Dextrose Agar and Plate Count Agar (VWR, Italy), respectively. After sampling, all the plates were incubated (molds at 25 °C for 10 days; total microbial counts at 22 °C for 72 h; total microbial counts at 37 °C for 48 h) as described elsewhere [28]. 

Following the incubation time, the molds were macroscopically and microscopically examined to detect the presence of *Aspergillus* spp, and further molds as described elsewhere [29].

### HVAC System Disinfection

The disinfection procedures of the HVAC systems were performed in both hospitals after the first air and surface samples were taken. A mixture of hydrogen peroxide (12%) with silver ions (10 mg/L) was aerosolized into all the air sampling points. The disinfectant was applied for at least 90 min contact time in order to cover the entire RDW surface and channels. All the treatments were applied to the HVAC systems in “at rest” conditions. Moreover, the room vents were temporarily sealed, avoiding the exposure of patients and healthcare personnel to the chemical compound. 

### Statistical Analysis

The Kolmogorov–Smirnov test was performed to verify the normality of the distributions. For each HVAC system, the Kruskall–Wallis test and the Dunn’s test were used to evaluate the reduction of the total microbial counts at 22 and 37 °C and the mold levels after disinfection. To assess the type of microbial contamination (environmental and mesophiles bacteria), we also compared the total microbial counts at 22 and 37 °C. The power tests were carried out to estimate the sample sizes. The 1-beta values of the significant variables were >0.8, proving acceptable sample sizes. The statistical analysis was fulfilled using the IBM SPSS software package, version 17.0.1.

## 5. Conclusions

Our study is the first to highlight the need for new procedures for the disinfection of complex rotating dehumidification wheels, planned and installed in the HVAC systems. Despite the fact that the literature data do not report a disinfection strategy for the internal complex devices of the HVAC systems, we recommend the application of a safety plan for indoor air quality, preventing the infectious risk from airborne pathogens in HVAC systems of hospital’s surgical and intensive care areas.

## Figures and Tables

**Figure 1 pathogens-08-00008-f001:**
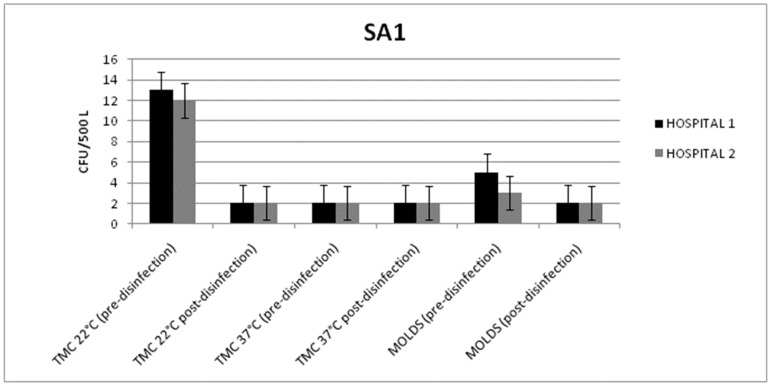
Mean values of the total microbial count (TMC) at 22/37 °C and the mold count detected in the air samples (before and after disinfection) from the heating, ventilation and air conditioning (HVAC) system of surgical area 1 (SA1) in Hospitals 1 and 2.

**Figure 2 pathogens-08-00008-f002:**
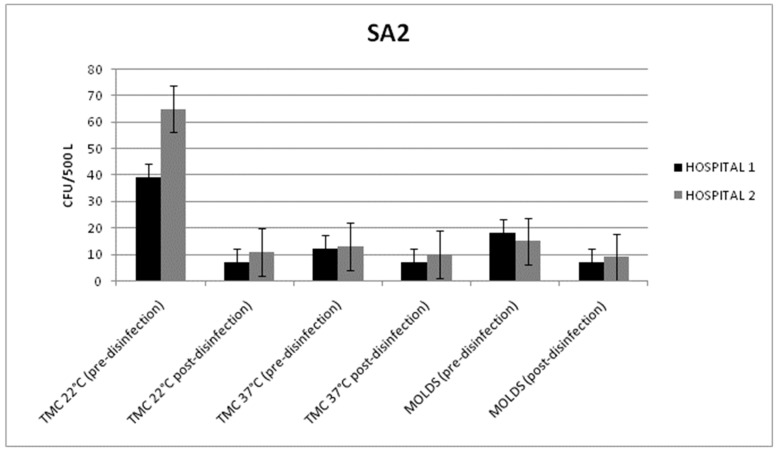
Mean values of the total microbial count (TMC) at 22/37 °C and the mold count detected in the air samples (before and after disinfection) from the HVAC systems of surgical area 2 (SA2) in Hospitals 1 and 2.

**Figure 3 pathogens-08-00008-f003:**
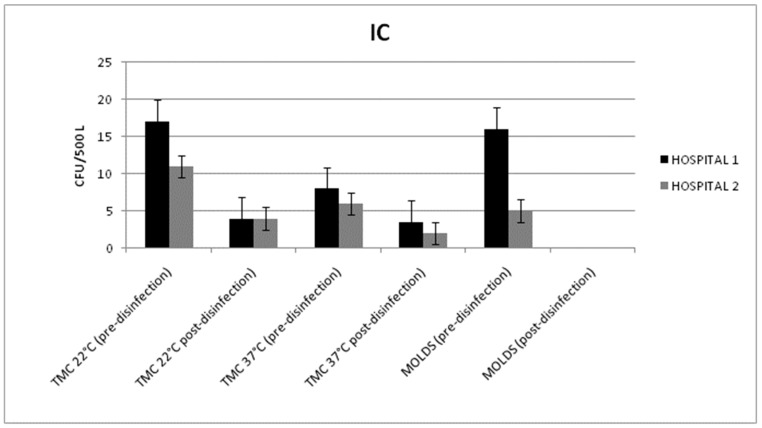
Mean values of the total microbial count (TMC) at 22/37 °C and the mold counts detected in the air samples (before and after disinfection) from the HVAC systems of the intensive care (IC) units in Hospitals 1 and 2.

**Figure 4 pathogens-08-00008-f004:**
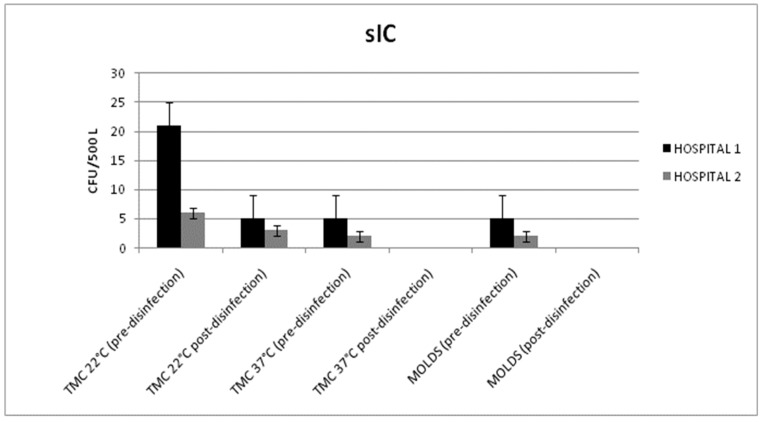
Mean values of the total microbial count (TMC) at 22/37 °C and the mold counts detected in the air samples (before and after disinfection) from the HVAC systems of the sub-intensive care (sIC) units in Hospitals 1 and 2.

**Figure 5 pathogens-08-00008-f005:**
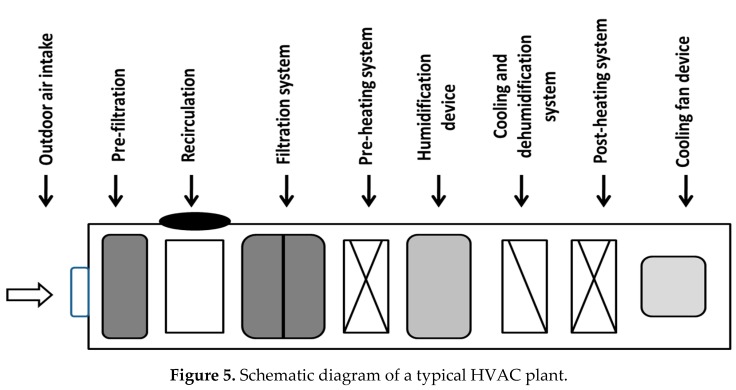
Schematic diagram of a typical HVAC plant.

**Figure 6 pathogens-08-00008-f006:**
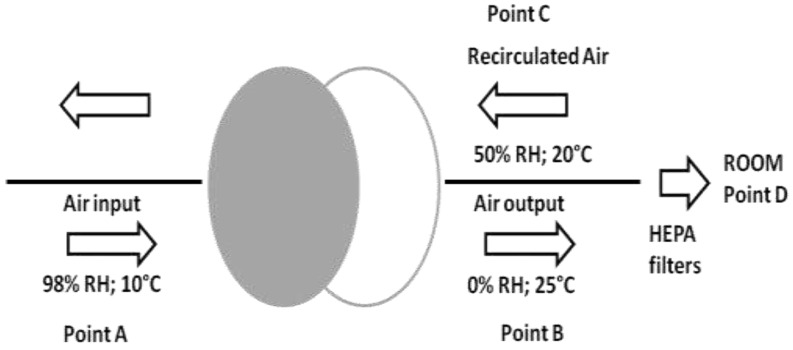
Process of dehumidification and the recirculation of air obtained from the rotating dehumidification wheels (RDWs). (RH = Relative Humidity).

**Table 1 pathogens-08-00008-t001:** Air and surface sampling protocols applied to the HVAC systems of Hospitals 1 and 2 before and after rotating dehumidification wheel (RDW) disinfection. (SA1 and SA2: surgery areas; IC and sIC: intensive and sub-intensive areas).

Hospital	Months of Samplings	Hvac Systems	Sampling Conditions	Matrix
HOSPITAL 1	September–December 2016	SA1 SA2 IC sIC	First sampling with RDW in operation	
	May–July 2017		Second sampling with RDW in operation (24 h after the disinfection)	Air and surfaces
HOSPITAL 2	April–May 2018	SA1 SA2 IC sIC	First sampling with RDW in operation	
	July 2018		Second sampling with RDW in operation (24 h after the disinfection)	Air and surfaces

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
