# Peer review of "Microbiological Air Quality in Heating, Ventilation and Air Conditioning Systems of Surgical and Intensive Care Areas: The Application of a Disinfection Procedure for Dehumidification Devices"

_pathogens, 2019, doi:10.3390/pathogens8010008_

Round 1

Reviewer 1 Report

Airborne pathogens residing in Heating, Ventilation and Air Control (HVAC) systems are a source of nosocomial infections in hospitals. HVACs installed with Rotating Dehumidification Wheels (RDW) further complicates the cleaning process. This study applied a disinfection strategy (employing hydrogen peroxide and silver ions) to RDWs installed in HVACs at two hospitals in Italy and evaluated its efficacy in reducing microbial load. RDWs sampled were associated with two surgical areas, an intensive care unit and a sub-intensive care unit. By measuring the CFUs of total microbes and fungi before and after disinfection, the authors show that their procedure reduced the total microbes and fungi in the RDWs examined. Importantly, the pathogenic filamentous fungus Aspergillus fumigatus was detected in one of the RDWs and was successfully eliminated following treatment, further highlighting the usefulness of the system in eliminating thermotolerant pathogens.

The authors should address the following minor comments prior to publication:

Line 57: “microbial counts at 22 and 37°C ranged from 2 to 13 CFU/500L….” - Here the order is flipped. It should be 13 and 2 CFU/500L respectively.

Line 60: It is not clear what data sets the p-value refers to? Here, the authors state that no significant reduction in CFUs were observed post-treatment, however, the data for TMC at 22°C shows a reduction. The authors should clarify this and present p-values.

Can the authors comment on why TMC at room temperature is significantly higher that at 37°C in all cases examined?

Can the authors determine what fungal genus were commonly present in their analyses? This information would be valuable to the community.

Can the authors comment on TMCs at 37 following treatment, especially in sIC (fig. 4) when there’s a reduction? Is this statistically significant?

It is a bit confusing when the CFU is expressed as pure numbers in the text versus as e powers in the graph. The authors should be consistent.

Figs. 5 and 6 are identical. They must be combined.

Since the journal is “Pathogens” it is also important that the authors show the data pertaining to Aspergillus fumigatus and not just state it in the text. Can they show an image of the isolate and present CFUs as a graph?

Author Response

Airborne pathogens residing in Heating, Ventilation and Air Control (HVAC) systems are a source of nosocomial infections in hospitals. HVACs installed with Rotating Dehumidification Wheels (RDW) further complicates the cleaning process. This study applied a disinfection strategy (employing hydrogen peroxide and silver ions) to RDWs installed in HVACs at two hospitals in Italy and evaluated its efficacy in reducing microbial load. RDWs sampled were associated with two surgical areas, an intensive care unit and a sub-intensive care unit. By measuring the CFUs of total microbes and fungi before and after disinfection, the authors show that their procedure reduced the total microbes and fungi in the RDWs examined. Importantly, the pathogenic filamentous fungus Aspergillus fumigatus was detected in one of the RDWs and was successfully eliminated following treatment, further highlighting the usefulness of the system in eliminating thermotolerant pathogens.

The authors should address the following minor comments prior to publication:

Line 57: “microbial counts at 22 and 37°C ranged from 2 to 13 CFU/500L….” - Here the order is flipped. It should be 13 and 2 CFU/500L respectively.

This order has been changed.

Line 60: It is not clear what data sets the p-value refers to? Here, the authors state that no significant reduction in CFUs were observed post-treatment, however, the data for TMC at 22°C shows a reduction. The authors should clarify this and present p-values.

Statement has been modified as requested and p-values >0.05 have been added in the text.

Can the authors comment on why TMC at room temperature is significantly higher that at 37°C in all cases examined?

Further statistical results assert that TMC at 22°C is significantly higher that at 37°C only in all HVAC systems sampled before the disinfection. This data assert the prevalence of environmental bacteria contamination. p-values have been calculated and this comment has been added in Discussion and Methods section.

Can the authors determine what fungal genus were commonly present in their analyses? This information would be valuable to the community.

Some fungal genus has been specified in Discussion section.

Can the authors comment on TMCs at 37 following treatment, especially in sIC (fig. 4) when there’s a reduction? Is this statistically significant?

In Discussion section we added the following statement “In fact, considering the low level of mesophilic bacteria contamination we have not observed statistically significant reduction of total microbial counts at the 37°C after disinfections”. A comment of TMCs at 37°C following sIC disinfection has been added in Results.

It is a bit confusing when the CFU is expressed as pure numbers in the text versus as e powers in the graph. The authors should be consistent.

All graphs have been modified as suggest by you.

Figs. 5 and 6 are identical. They must be combined.

It was a mistake. Figure 5 (changed in Figure 6) has been added.

Since the journal is “Pathogens” it is also important that the authors show the data pertaining to Aspergillus fumigatus and not just state it in the text. Can they show an image of the isolate and present CFUs as a graph?

Microscopy image has been added as Figure 4. It s not possible add a graph because Aspergillus fumigatus was detected and isolated in just one sample.

Reviewer 2 Report

The manuscript "Microbiological air quality in heating, ventilation and air conditioning systems of surgical and intensive care areas: application of a disinfection procedure for the dehumidification devices" treats the problem of  healthcare correlated infections resulting increasing. However the work needs major revisions to be improved.

In section 4 “Materials and methods”, lines 178-179, is not clear microbiological analysis for the research of fungi and total microbial counts at 22° and 37° C. More information must be reported: 5 days are enough for fungi growth? According to which method? Why was performed the total microbial count at 22 ° and 37 ° C and not only 37°C?

Is the incubation for 24 h or 48 h sufficient for microbial count at 37°C? Describe better the identification of fungi. It is unclear whether only an automated system such as vitek 2 has been utilized, or a micro and macroscopic identification has been performed, as reported by other authors, e.g. Caggiano et al., 2014.

Caggiano G, Napoli C, Coretti C, Lovero G, Scarafile G, De Giglio O, Montagna MT. Mold contamination in a controlled hospital environment: a 3-year surveillance in southern Italy. BMC Infect Dis. 2014 Nov 15;14:595. doi: 10.1186/s12879-014-0595-z.

Moreover, from the reference it seems that the methods regarding microbiological investigations are described in the Guidelines Ispesl 2009 but it is not so: the Guidelines Ispesl 2009 does not report the cultural medium and the incubation temperature for microbiological analysis of air. If the methods take up the literature, it must be referred to this.  

           .            

Author Response

The manuscript "Microbiological air quality in heating, ventilation and air conditioning systems of surgical and intensive care areas: application of a disinfection procedure for the dehumidification devices" treats the problem of  healthcare correlated infections resulting increasing. However the work needs major revisions to be improved.

In section 4 “Materials and methods”, lines 178-179, is not clear microbiological analysis for the research of fungi and total microbial counts at 22° and 37° C.

According to you the Methods section has been revised and the question points have been argued.

Why was performed the total microbial count at 22 ° and 37 ° C and not only 37°C?

In the text we have added a statement about the assessment of different type of bacteria contamination (mesophiles and not). Moreover, in this way is possible to assess the efficacy of hydrogen peroxide in reduction of both total microbial counts.    

Is the incubation for 24 h or 48 h sufficient for microbial count at 37°C?

“24 h” was a mistake. Incubation time for TMCs at 37°C was 48 h, as reported by the cited reference.

5 days are enough for fungi growth? According to which method?  Describe better the identification of fungi. It is unclear whether only an automated system such as vitek 2 has been utilized, or a micro and macroscopic identification has been performed, as reported by other authors, e.g. Caggiano et al., 2014.

According to you, all microbiological analysis about fungi research has been re-written in more specific form. Incubation time for fungi growth was 10 days, but in cases of high fungal contamination, times have been shorter (5-7 days).

Moreover, from the reference it seems that the methods regarding microbiological investigations are described in the Guidelines Ispesl 2009 but it is not so: the Guidelines Ispesl 2009 does not report the cultural medium and the incubation temperature for microbiological analysis of air. If the methods take up the literature, it must be referred to this.  

According to you. More technical references methods have been added and bibliography has been updated.

Round 2

Reviewer 2 Report

The manuscript "Microbiological air quality in heating, ventilation and air conditioning systems of surgical and intensive care areas: application of a disinfection procedure for the dehumidification devices" has improved in this new version. However it needs further major revisions before being accepted.

·         Figure 4. From the microscopic image it does not seem to be Aspergillus fumigatus. Can the author also report the image of the cultural analysis?

·         Linee 139. In Discussion reference is made to Cladosporium and Penicillium, but they do not appear in the results. What genus or species of molds have been isolated beyond Aspergillus fumigatus?

·         In the text the term “fungi” is always used. In fact it would be advisable in some cases to replace “fungi” with the more appropriate term “filamentous fungi” or “molds”. For example linee 204 "Following the incubation time fungi ...." add “filamentous” fungi or replace with “molds”.

·         Linee 205. “…further pathogen mold as described elsewhere”. Aspergillus fumigatus is an opportunistic pathogen so I would eliminate the word "pathogen".

Author Response

The manuscript "Microbiological air quality in heating, ventilation and air conditioning systems of surgical and intensive care areas: application of a disinfection procedure for the dehumidification devices" has improved in this new version. However it needs further major revisions before being accepted.

Figure 4. From the microscopic image it does not seem to be Aspergillus fumigatus. Can the author also report the image of the cultural analysis?

Unfortunately we don’t have images of Aspergillus fumigatus cultural analysis. Considering that the microscopic image may be hard to define it has been removed. 

 Linee 139. In Discussion reference is made to Cladosporium and Penicillium, but they do not appear in the results. What genus or species of molds have been isolated beyond Aspergillus fumigatus?

During the whole period of study further filamentous fungi have been isolated in sporadic cases. Considering the low pathogenicity of these microorganisms our research was not depth in results section. However a statement (data not shown) has been added in the text.

In the text the term “fungi” is always used. In fact it would be advisable in some cases to replace “fungi” with the more appropriate term “filamentous fungi” or “molds”. For example linee 204 "Following the incubation time fungi ...." add “filamentous” fungi or replace with “molds”.

Corrections have been applied in the text and figures.

Linee 205. “…further pathogen mold as described elsewhere”. Aspergillus fumigatus is an opportunistic pathogen so I would eliminate the word "pathogen".

Correction has been applied

Round 3

Reviewer 2 Report

Now this manuscript can be published